# *Ulva compressa* from Copper-Polluted Sites Exhibits Intracellular Copper Accumulation, Increased Expression of Metallothioneins and Copper-Containing Nanoparticles in Chloroplasts

**DOI:** 10.3390/ijms221910531

**Published:** 2021-09-29

**Authors:** Daniela Espinoza, Alberto González, Jaime Pizarro, Rodrigo Segura, Daniel Laporte, Fernanda Rodríguez-Rojas, Claudio A. Sáez, Alejandra Moenne

**Affiliations:** 1Laboratory of Marine Biotecnology, Faculty of Chemistry and Biology, University of Santiago of Chile, Santiago 9170022, Chile; daniela.espinoza@usach.cl (D.E.); alberto.gonzalezfi@usach.cl (A.G.); 2Laboratory of Inorganic Chemistry, Faculty of Chemistry and Biology, University of Santiago of Chile, Santiago 9170022, Chile; jaime.pizarror@usach.cl (J.P.); rodrigo.segura@usach.cl (R.S.); 3Laboratorio Multidisciplinario, Instituto de Ciencias Biomédicas, Universidad Autónoma de Chile, Talca 3467987, Chile; daniel.laporte.p@gmail.com; 4Laboratory of Aquatic Environmental Research, Hub Ambiental UPLA, Centro de Estudios Avanzados, Universidad de Playa Ancha, Valparaíso 2340000, Chile; fernanda.rodriguez@upla.cl (F.R.-R.); claudio.saez@upla.cl (C.A.S.); 5Departamento de Ciencias del Mar y Biología Aplicada, Universidad de Alicante, 03690 Alicante, Spain

**Keywords:** copper, glutathione, marine alga, metallothionein, phytochelatins, *Ulva compressa*

## Abstract

In order to analyze the mechanisms involved in copper accumulation in *Ulva compressa*, algae were collected at control sites of central and northern Chile, and at two copper-polluted sites of northern Chile. The level of intracellular copper, reduced glutathione (GSH), phytochelatins (PCs), PC2 and PC4, and transcripts encoding metallothioneins (MTs) of *U. compressa*, *UcMT1*, *UcMT2* and *UcMT3*, were determined. Algae of control sites contained around 20 μg of copper g^−1^ of dry tissue (DT) whereas algae of copper-polluted sites contained 260 and 272 μg of copper g^−1^ of DT. Algae of control sites and copper-polluted sites did not show detectable amounts of GSH, the level of PC2 did not change among sites whereas PC4 was increased in one of the copper-polluted sites. The level of transcripts of *UcMT1* and *UcMT2* were increased in algae of copper-polluted sites, but the level of *UcMT3* did not change. Algae of a control site and a copper-polluted site were visualized by transmission electron microscopy (TEM) and the existence of copper in electrodense particles was analyzed using energy dispersive x-ray spectroscopy (EDXS). Algae of copper-polluted sites showed electrodense nanoparticles containing copper in the chloroplasts, whereas algae of control sites did not. Algae of a control site, Cachagua, were cultivated without copper (control) and with 10 μM copper for 5 days and they were analyzed by TEM-EDXS. Algae cultivated with copper showed copper-containing nanoparticles in the chloroplast whereas control algae did not. Thus, *U. compressa* from copper-polluted sites exhibits intracellular copper accumulation, an increase in the level of PC4 and expression of *UcMTs*, and the accumulation of copper-containing particles in chloroplasts.

## 1. Introduction

Heavy metals are classified as essential when they act as cofactors of proteins and enzymes [1,2]. Essential heavy metals are required only in trace amounts since in excess they induce the production of reactive oxygen species such (ROS) as superoxide anions and hydrogen peroxide [3]. ROS can directly oxidize biomolecules such as fatty acids, proteins and nucleic acids, leading eventually to cell death [1,2]. There are also non-essential heavy metals such as cadmium, lead, mercury, chromium and others and these non-essential heavy metals are toxic even in trace amounts [1,2]. The principal mechanism to buffer oxidative stress induced by heavy metals is the activation of antioxidant enzymes, such as superoxide dismutase, which dismutate superoxide anions to hydrogen peroxide; catalase (CAT), ascorbate peroxidase (AP), glutathione peroxidase (GP) and peroxiredoxin (PR) that convert hydrogen peroxide into oxygen and water [2,3]. In addition, ROS can be directly buffered by antioxidant compounds such as ascorbate (ASC) and glutathione (GSH) and these antioxidant compounds are also the substrates of AP and GP. Moreover, glutathione reductase (GR) uses NADPH as reducing power [3].

Another mechanism to buffer oxidative stress induced by heavy metals is the synthesis cysteine-rich peptides and/or proteins that sequester these metals [4]. The sequestering of heavy metals can be achieved by GSH, a tripeptide constituted by glutamate, cysteine and glycine, and/or by phytochelatins (PCs) that are formed by condensed units of GSH (*n* = 2–12) synthesized by the enzyme phytochelatin synthase [4]. Proteins that sequester heavy metals are metallothioneins (MTs) which are small proteins (<10 kDa) containing a high percentage of cysteine residues (20–30% of amino acids) and hydrophobic amino acids such as glycine and alanine, and a low percentage of aromatic amino acids [4]. MTs can sequester divalent and monovalent cations such as copper, zinc, cadmium, lead, mercury and silver [4]. MTs are present in cyanobacteria, protists, fungi, nematodes, animals, plants and algae [5,6,7]. In animals, there are four MTs, in fish there are two MTs, in invertebrates there are mainly two MTs and in yeast there are two MTs [5]. In plants, there are mainly six MTs as it has been shown in *Arabidopsis thaliana* and *Populus trichocarpa x deltoides* [7]. In marine brown and red macroalgae there is only one MT, and until recently the only cloned and expressed MT was *Fucus vesiculosus* MT [7,8]. However, it has been recently shown that the green macroalga *Ulva compressa* contains at least three MTs that allow the accumulation of copper and zinc when expressed in bacteria [9].

The marine alga *U. compressa* is a cosmopolitan species that thrives in heavy metal-polluted sites in northern Chile and all over the world [10,11]. *U. compressa* can accumulate several heavy metals and it has been used a heavy metal bio-monitoring species in different places of the world [10,11]. It was initially shown that algae of the copper-polluted site Caleta Palito, which receives effluents from the copper mine El Salvador and is located in Chañaral Bay in northern Chile, accumulate copper in its tissue [11]. In addition, algae of copper-polluted sites showed the activation of AP antioxidant enzyme and the synthesis of ASC that was accumulated as DHA [11]. Interestingly, the levels of DHA were linearly correlated with the amount of copper in seawater [11]. Seawater of Caleta Palito contained 20 µg L^−1^ of copper and *U. compressa* contained 750 µg g^−1^ of dry tissue (DT); whereas seawater of the control site of central Chile, Cachagua, contained 2 µg L^−1^ of copper and algae contained 17 µg g^−1^ of DT [11]. Afterwards, the alga was cultivated with a sub-lethal concentration of copper (10 µM) for 7 days and showed an increase in SOD, AP and GR activities [12,13] as well as the synthesis of ASC, GSH and PCs [14].

It has been recently shown that the alga cultivated with increasing concentrations of copper in seawater corresponding to 2.5, 5, 7.5 and 10 µM for 0 to 12 days displayed an accumulation of intracellular copper that linearly correlated with increasing concentrations of copper [15]. The amounts of GSH increased mainly with lower concentrations of copper (2.5 and 5 µM) whereas PCs increased with higher concentrations of copper (7.5 and 10 µM) from day 1 to 12 [15]. Moreover, the level of transcripts encoding three MTs increased mainly with higher concentrations of copper from day 3 to 12 [15]. Moreover, it was shown that the alga cultivated in synthetic seawater with 10 µM of copper for 5 days, and then in seawater without copper for 3 days, displayed a decrease in intracellular copper and a concomitant increase in copper level in the culture medium [15]. Interestingly, an equimolar amount of GSH was detected in the extracellular medium suggesting that GSH may participate in copper extrusion [15]. Furthermore, transcripts encoding three MTs from *U. compressa,* designated *UcMT1*, *UcMT2* and *UcMT3*, were cloned, sequenced and expressed in *E. coli* [9]. The overexpression of the transcripts *UcMT1*, *UcMT2* and *UcMT3* allowed the accumulation of copper and zinc in vivo [9]. Overexpression of *UcMT1* and *UcMT2* allowed a higher accumulation of copper compared with *UcMT3* [9], suggesting that these MTs could mediate copper accumulation in *U. compressa*.

In order to analyze the mechanisms involved in copper accumulation and to determine the cellular compartment in which copper is accumulated, *U. compressa* and seawater were collected at control sites of central and northern Chile and at copper-polluted sites of northern Chile. The levels of copper in seawater, the amount of intracellular copper, the levels of GSH and PCs and the levels of *UcMT* transcripts were analyzed in algae collected in control and copper-polluted sites. In addition, algae of a control and a copper-polluted site were visualized using TEM and the presence of copper was analyzed using EDXS. Moreover, algae from a control site were cultivated with 10 µM copper for 5 days and the presence of copper-containing particles was analyzed by TEM-EDXS.

## 2. Results

### 2.1. Quantification of Copper in Seawater of Control Sites and Copper-Polluted Sites

Seawater was collected at control sites located in northern Chile, Caleta Zenteno, Bahía Inglesa and Chañaral de Aceituno, and in central Chile, Cachagua and Ventanas. Additionally, algae were collected at copper-polluted sites located in northern Chile, Caleta Palito and Chañaral Sur that is located near Caleta Palito (Figure 1). The levels of copper, cadmium and lead were analyzed in seawater of control sites and copper-polluted sites using anodic stripping voltammetry. Heavy metals were undetectable at control sites (<0.19 µg L^−1^) whereas only copper was detected at copper-polluted sites with concentrations of 1.97 µg L^−1^ in Caleta Palito and 3.62 µg L^−1^ Chañaral Sur (Table 1). Thus, Caleta Palito and Chañaral Sur are two copper-polluted sites of northern Chile.

### 2.2. Quantification of Copper in Algae of Control Sites and Copper-Polluted Sites

The level of intracellular copper in *U. compressa* collected at control sites and copper-polluted sites was analyzed using atomic absorbance spectroscopy. The level of intracellular copper in algae of control sites ranged around 20 µg g^−1^ of dry tissue (DT). In algae of copper-polluted sites the level of intracellular copper was 276 µg g^−1^ of DT in Caleta Palito, and 260 µg g^−1^ of DT in Chañaral Sur and these levels were not significantly different among them (Figure 2A). Thus, *U. compressa* from copper-polluted sites accumulated similar intracellular levels of copper, even if concentrations of copper in seawater were different (see above).

Besides, the level of cadmium in algae of control sites Bahia Inglesa, Chañaral de Aceituno and Caleta Palito was 1.2, 1.1 and 1.1 µg g^−1^ of DT, respectively. The level of cadmium in Caleta Zenteno was 8.2 µg g^−1^ of DT (Figure 2B), even if no cadmium was detected in seawater of Caleta Zenteno. Thus, algae at control sites did not show accumulation of copper or cadmium, except those of Caleta Zenteno that accumulated cadmium.

### 2.3. Quantification of GSH and PCs in Algae of Control Sites and Copper-Polluted Sites

The level of GSH was undetectable in algae from control sites and copper-polluted sites (data not shown). The level of PC2 in algae of control sites was 1.45 to 2.8 nmoles g^−1^ of FT and the level of PC2 was not significantly higher at copper-polluted sites compared to control sites (Figure 3A). In addition, the level of PC4 at control sites ranged from 1.4 to 1.8 nmoles g^−1^ of fresh tissue (FT) whereas the level of PC4 at the copper-polluted site Caleta Palito was 2.4 nmoles g^−1^ of FT, which is markedly higher than PC4 in algae of control sites. In algae of Chañaral Sur, the level of PC4 was 2.1 nmoles g^−1^ of FT, which was not significantly different from control sites (Figure 3B). Thus, PC4 was only increased in algae from Caleta Palito, and not in algae from Chañaral Sur.

### 2.4. Quantification of Transcripts UcMT1, UcMT2 and UcMT3 in Algae of Control Sites and Copper-Polluted Sites

The relative level of transcripts encoding *UcMT1* were significantly increased in algae of copper-polluted sites, Caleta Palito and Chañaral Sur, showing an increase of 1.5 and 4.9 times, respectively, whereas in algae of control sites it only increased 0.2 times (Figure 4A). In addition, the level of transcripts encoding *UcMT2* increased in Caleta Palito and Chañaral Sur in 4.4 and 2.8 times, respectively (Figure 4B). The level of transcripts encoding *UcMT2* was also increased (1.9 times) in algae of the control site Ventanas, but not in other control sites (Figure 4B). Moreover, the level of transcripts encoding *UcMT3* were increased in the control site Caleta Zenteno, in 0.5 times, but not in algae copper-polluted sites (Figure 4C). Thus, transcripts encoding *UcMT1* and *UcMT2*, but not UcMT3, were increased in algae of copper-polluted sites.

### 2.5. Copper-Containing Nanoparticles in Algae of a Copper-Polluted Site, but Not in Algae of a Control Site

Algae of Caleta Palito, a copper-polluted site, and Cachagua, a control site, were visualized by TEM and electrodense nanoparticles were analyzed to detect copper by EDXS. Algae from a control site, Cachagua, showed electrodense nanoparticles (Figure 5A–D) but these nanoparticles did not contain copper (Figure 5D). Algae of the copper-polluted site, Caleta Palito, showed electrodense nanoparticles of around 20–50 nm that contained copper (Figure 5E–H and Appendix A) and these nanoparticles were located in the chloroplast (Figure 5H).

### 2.6. Copper-Containing Nanoparticles in Algae of a Control Site Cultivated with Copper, but Not in Algae of the Control Site Cultivated without Copper

Algae of a control site, Cachagua, were cultivated in seawater without copper addition (control) and with 10 µM of copper for 5 days. Algae cultivated in control condition showed electrodense nanoparticles, but these nanoparticles did not contain copper (Figure 6A–D). In contrast, algae cultivated with copper showed electrodense nanoparticles containing copper and they were located in the chloroplast (Figure 6E–H).

## 3. Discussion

In this work, we showed that the marine alga *U. compressa* collected at copper-polluted sites accumulates intracellular copper and these levels were not significantly different among copper-polluted sites. However, the concentrations of copper in seawater at copper-polluted sites were different since at Caleta Palito it was 1.97 µg L^−1^ (0.03 µM) and ay Chañaral Sur it was 3.62 µg L^−1^ (0.1 µM). This indicates that algae chronically exposed to copper accumulate the metal. However, the level of intracellular copper did not accurately reflect the concentration of this metal in seawater since in both cases the alga showed an intracellular concentration of copper around 260–270 µg g^−1^ of DT. These results differ from those obtained with the alga cultivated with increasing concentrations of copper corresponding to 2.5, 5, 7.5 and 10 µM for 0 to 12 days that showed a linear accumulation of copper with increasing time [15]. The alga chronically exposed to different concentrations of copper 0.03 and 0.1 µM accumulates similar levels of intracellular copper (this work), suggesting that the alga chronically exposed to copper may continuously extrude copper to the extracellular medium, probably using GSH, as it was previously shown [15]. On the other hand, seawater at Caleta Palito in 2002 presented a concentration of copper of 20 µg L^−1^ [11] whereas in 2019 this concentration was 1.97 µg L^−1^ indicating that mitigating measures implemented by the El Salvador mine in the last 15 years has led to an improvement in sustainability of copper extraction in northern Chile.

Interestingly, the level of GSH was undetectable in algae of copper-polluted sites. This contrasts with the fact that the alga exposed to 10 µM for 7 d synthesizes GSH due to an increased expression of enzymes involved in its production [14]. It is possible that synthesized GSH may be required for copper extrusion and may inhibit the recapture of copper by the alga. In this sense it has been shown that microalgae such as *Emiliana huxleyi* and *Phaeodactylum tricornutum* can release GSH in response to the addition of nanomolar concentrations of copper in seawater, probably to inhibit copper entry into the cells [16,17,18]. Besides, green, red and brown macroalgae such as *Porphyra* sp. *Ulva* sp. and *Fucus vesiculosus* can also release cysteine, GSH and/or phenolic compounds such as phlorotannins in response to copper addition in seawater in order to inhibit copper entry into cells [19,20,21]. Therefore, GSH may be required for the release of copper and to inhibit its recapture in *U. compressa* chronically exposed to copper excess.

In addition, the level of PC2 was not increased in algae of copper-polluted sites whereas the level of PC4 was significantly increased in algae from Caleta Palito, but not in algae from Chañaral Sur. These results differ from those obtained with the alga cultivated with increasing concentrations of copper for 0 to 12 days where PCs and GSH, expressed as GSH equivalents, correlated with the level of intracellular copper [15]. Moreover, the levels of PC2 and PC4 in algae of copper polluted sites were extremely low compared to those in the alga cultivated with copper for 12 days [15]. The low level of PCs in algae chronically exposed to copper excess may result from the use of GSH to release copper to the extracellular medium and because GSH is used to synthesize PCs but, it is not excluded that PCs can be used to accumulate copper in the chloroplast (see below).

On the other hand, the level of transcripts *UcMT1* and *UcMT2* increased in algae from copper-polluted sites compared with those from control sites, but not those of *UcMT3*. In this sense, it has been shown that cloned *UcMT1* and *UcMT2* expressed in bacteria allowed a higher accumulation of copper and zinc compared to control [9], suggesting that these proteins may participate in copper accumulation in the alga. Until now, it has not been described a metallothionein directed to the chloroplast allowing copper or other metal accumulation. Thus, it is unlikely, but not excluded, that copper accumulation in the chloroplasts may be mediated by UcMTs. In addition, there was an increase in the level of *UcMT3* transcripts in algae from Bahía Inglesa, a site of northern Chile, and an increase in the level of *UcMT2* in algae from Ventanas, a site of central Chile. The latter may be due to the presence of other heavy metals such as V, Mo, Fe, Ni and Cd in seawater due to upwellings that carried metals from the bottom of the ocean as it has been observed in seawater of the Mejillones Bay located in northern Chile [22]. This suggests that expression of UcMTs may increase in response to other heavy metals different from Cu and Zn [9].

Furthermore, algae collected at a copper-polluted site, Caleta Palito, were analyzed by TEM-EDXS and showed electrodense nanoparticles containing copper. In contrast, algae from a control site, Cachagua, showed electrodense nanoparticles but they did not contain copper. In addition, algae from a control site, Cachagua, cultivated with 10 µM of copper for 5 d also showed copper-containing electrodense nanoparticles, and algae cultivated without copper addition did not show copper-containing particles. Moreover, copper-containing nanoparticles were located in the chloroplast of algal cells. In this sense, it has been previously shown that the green macroalga *U. flexuosa* accumulates copper in its tissue, mainly in the chloroplast, the vacuole, and in bacteria bound to the cell wall [23]. Moreover, it has been shown that cadmium and copper can be transported and accumulated in the vacuole of *Schizosaccharomyces pombe*, and in chloroplasts of *E. gracilis* and different plants, complexed with PCs and sulfide, since the complex with PCs and sulfide is more stable than the complex formed with only by PCs [24,25,26,27,28]. It is interesting to point out that the production of sulfide occurs in chloroplasts since the enzyme sulfite reductase, that transforms sulfite to sulfide, and D/L-cysteine desulfurylase, that produces sulfide from cysteine, are located exclusively in plant chloroplasts [29]. Thus, it is possible that in *U. compressa* copper is accumulated in chloroplasts bound to PCs and/or to sulfide.

In conclusion, *U. compressa* from copper-polluted sites showed the accumulation of copper, the increase in PC4 and *UcMT1–2* transcript levels. Algae from a copper-polluted site and algae from a control site cultivated with copper showed electrodense nanoparticles containing copper that may have been formed by copper-bound PCs, sulfide and/or UcMTs and these particles were accumulated in chloroplasts.

## 4. Materials and Methods

### 4.1. Sampling of Seawater and U. compressa

Seawater (50 mL) and the alga *U. compressa* (10 g) were collected at potential copper-polluted sites in northern Chile corresponding to Caleta Palito and Chañaral Sur; at control sites of northern Chile corresponding to Zenteno, Chañaral de Aceituno and Bahía Inglesa; and at control sites of central Chile, Ventanas and Cachagua. Algae were transported to the laboratory in a cooler with ice and they were rinsed with filtered seawater obtained in Quintay, a pristine site of central Chile. Algae were cleaned manually and sonicated twice for 2 min in an ultrasound bath (HiLab Innovation Systems, model SK221 OHP, JiLin, China) in order to remove epiphytic bacteria. Algae were washed three times with 50 mM Tris-HCl (pH 7.8)–10 mM EDTA in order to remove copper ions from the cell wall [30]. The algae were maintained in aerated seawater for 4 days, previous to experimental procedures.

### 4.2. Quantification of Copper in Seawater

Seawater (50 mL) was obtained from control sites and copper-polluted sites and the levels of copper, cadmium and lead were determined by anodic stripping voltammetry, a method that shows a limit of detection of 0.19 µg L^−1^ [31]. The calibration curve was prepared with 1, 2, 5, 10 and 20 µg mL^−1^ of copper, cadmium and lead.

### 4.3. Quantification of Intracellular Copper in U. compressa

Algae (500 mg of dry tissue) were incubated in 5 mL of nitric acid 65% (ultrapure, Merck) and 2 mL of hydrogen peroxide (Merck, Burlington, MA, USA) in teflon vials at room temperature for 2 h, at 200 °C for 20 min and 1000 W, for 10 min at 200 °C and 1000 W and cooled at room temperature for 30 min. The solutions were diluted with 10 mL of ultrapure water. Quantification of intracellular copper was performed by atomic absorption spectroscopy model ICE 3300 (Thermo Scientific, Waltham, MA, USA) and for calibration curves concentrations of 0, 5, 10, 15, 20 µg mL^−1^ for copper; 0, 0.25, 0.5, 0.75, 1, 1.5 and 2 µg mL^−1^ for cadmium; and 0, 0.5, 0.75 and 1 µg mL^−1^ for lead were used.

### 4.4. Quantification of GSH and PCs in U. compressa

Algae (200 mg of fresh tissue) were frozen with liquid nitrogen and homogenized in a mortar using a pestle and 1.2 mL of 0.1% trifluoroacetic acid (TFA)/6.3 mM dietilentriamine pentacetic acid (DTPA) were added. The mixture was centrifuged at 14.000 rpm for 20 min and the supernatant was recovered. The supernatant was filtered through 0.22 µM PVDF membrane and an aliquot of 25 µL of the supernatant was mixed with 45 µL of 200 mM HEPES (pH 8)–6.3 mM DTPA and 1 µL of 25 mM monobromobimane and incubated at room temperature for 30 min, in darkness. The reaction was stopped by addition of 30 µL of 1 mM metasulphonic acid. The amount of GSH and PCs (PC2 and PC4) was determined by high performance liquid chromatography 1260 Infinity (Agilent Technologies, Santa Clara, CA, USA) and data was compiled using OpenLab CDS software (iCD System, Köln, Germany). An aliquot of 20 µL was separated on a reverse phase C-18 column (5 µM particle size, 4.6 mm internal diameter and 15 cm length) at 24 °C, eluted using solvent A (0.1% TFA in aqueous solution) and solvent B (100% acetonitrile) and a linear gradient of 10 min from 0–20%, 30 min from 20–35% and 10 min from 35–100% of solvent B, and a flow rate of 1 mL min^−1^. GSH, PC2 and PC4 were detected using a fluorescence detector at an excitation wavelength of 380 nm and an emission wavelength of 470 nm. Retention times of GSH, PC2 and PC4 were 9.25, 11.6 and 16.8 min, respectively. Pure GSH, PC2 and PC4 were dissolved in filtered water and used as standards and the calibration curves were prepared using PC2 and PC4 at concentrations from 0.8 to 50 µM.

### 4.5. Preparation of Total RNA and Quantification of UcMT Transcripts

Algae (100 mg of FT) were frozen with liquid nitrogen and pulverized in a mortar with a pestle. Total RNAs were extracted using FavorPrep Plant Total RNa kit (Favorgene, Wien, Austria) and quantified using an Infinite F500 reader (Tecan, Zurich, Switzerland). The relative levels of transcripts of *UcMT1* were determined using MT1-F: 5′CCAGTGCCAAACCGAAGATG3′ and MT1-R: 5′TGCTAGCAGGCACAGTCGTC3′ primers; transcripts of *UcMT2* were determined using the primers MT2-F: 5′AACTGCGACTGTGCTGACCA3′and MT2-R: 5′CAGTCTCGATGTGGCTCTTCT3′; transcripts of *UcMT3* were determined using the primers MT3-F: 5′GCAACCCTGAGACCTGCACT3′ and MT3-R: 5′CACAGTTGCATTCTGCGGTT3′. Transcripts of β*-tubulin*, the housekeeping transcript has been previously used in *U. compressa* treated with copper, were determined using the primers TUB-F: 5′TGCAACTTTTGTAGGCAACTC3′and TUB-R: 5′CAGTGAACTCCATCTCGTCC3′. Synthesis of cDNAs was performed using 1 µg of total RNA, 4 µL of iScript Reverse Transcription Supermix (BioRad, Santa Rosa, CA, USA), completing to a final volume of 20 µL with water treated with DEPC (water–DEPC) and incubated at 25 °C for 5 min, at 46 °C 20 min and at 95 °C 1 min. Amplification of UcMTs and β-tubulin transcripts was performed using 2 µL of cDNAs, 5 µL of SSO Advanced SYBR green Supermix (BioRad, Santa Rosa, CA, USA), 10 µM of each primer and completing to a final volume of 10 µL with water–DEPC. The amplification cycles (40) were performed at 95 °C for 40 s, at 55 °C for 10 s and 65 °C for 30 s using real time thermocycler Aria MX (Agilent Technologies, Santa Clara, CA, USA).

### 4.6. Preparation of Samples and Detection of Copper-Containing Nanoparticles

Algae (3 laminae) were introduced in 2 mL of 0.1 M cacodylate buffer (pH 7.2) containing 1% glutaraldehyde and stored at 4 °C. Algae were included in epoxide resin, stained with osmium and lead, and ultrathin sections (80 nm) were obtained using an ultramicrotome (Leica, Wetzlar, Germany). Samples were placed on gold grids and analyzed using a TEM of 300 keV model Titan (TemoFisher Scientific, Waltham, MA, USA) coupled to EDXS (Brucker, Coventry, UK) located at Center of Instrumentation, University of Granada, Spain.

### 4.7. Statistical Analyses

Significant differences were determined by two-way ANOVA followed by Tukey´s multiple comparison tests. Differences among mean values were considered to be significant at a probability of 5% (*p* < 0.05).

## Figures and Tables

**Figure 1 ijms-22-10531-f001:**
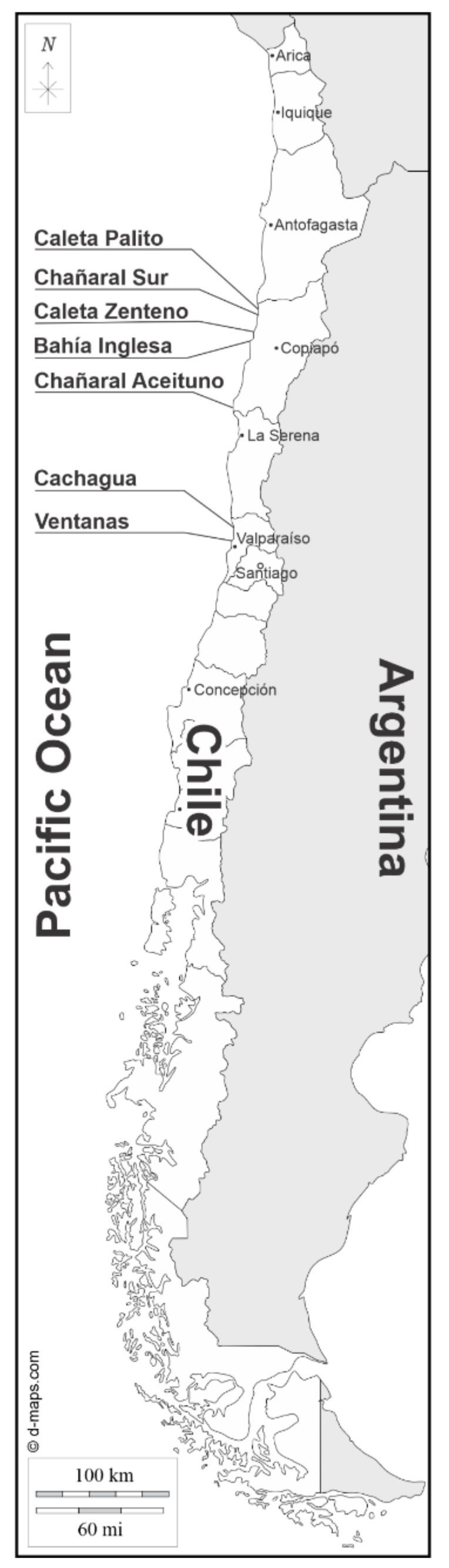
Map of Chile and location of copper-polluted sites in northern Chile, Caleta Palito and Chañaral Sur, control sites in northern Chile, Caleta Zenteno, Bahía Inglesa and Caleta Aceituno, and control sites in central Chile, Cachagua and Ventanas.

**Figure 2 ijms-22-10531-f002:**
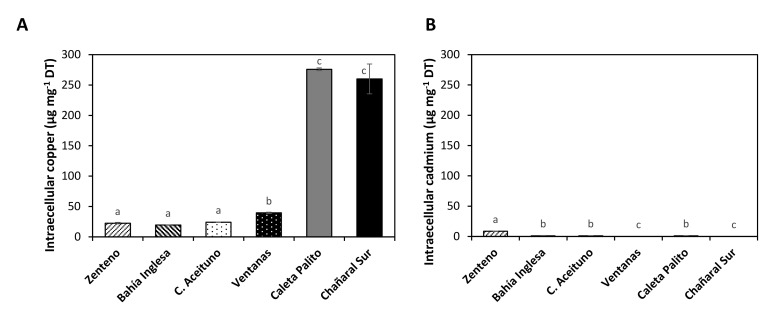
Level of intracellular copper (**A**) and cadmium (**B**) in algae of control sites and copper-polluted sites (**A**). The levels of intracellular copper and cadmium are expressed in micromoles per milligram of dry tissue (DT). Bars represent mean values of three independent replicates ± SD. Letters indicate significant differences (*p* < 0.05).

**Figure 3 ijms-22-10531-f003:**
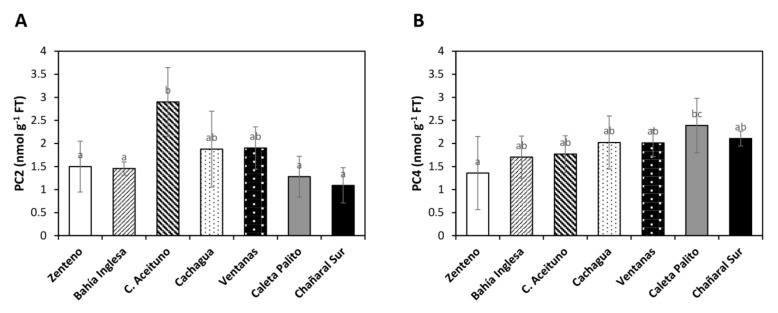
Level of phytochelatins, PC2 (**A**) and PC4 (**B**), in algae of control sites and copper-polluted sites. The levels of PC2 and PC4 are expressed in nanomoles per gram of fresh tissue (FT). Bars represent mean values of three independent replicates ± SD. Letters indicate significant differences (*p* < 0.05).

**Figure 4 ijms-22-10531-f004:**
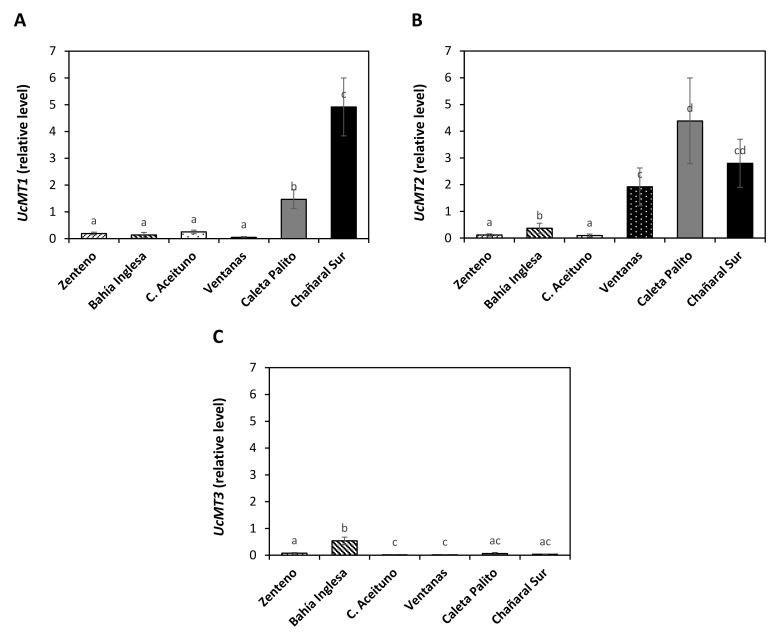
Relative level of transcripts encoding *UcMT1* (**A**), *UcMT2* (**B**) and *UcMT3* (**C**) in algae of control sites and copper-polluted sites. The levels of transcripts are expressed as 2^-ΔΔCT^. Bars represent mean values of three independent replicates ± SD. Letters indicate significant differences (*p* < 0.05).

**Figure 5 ijms-22-10531-f005:**
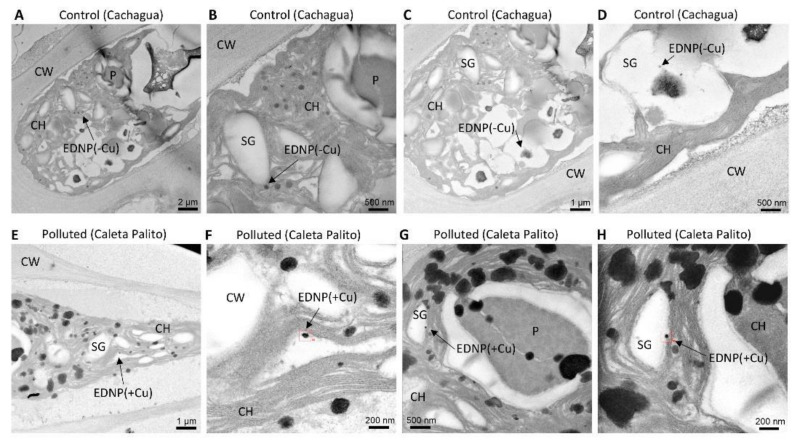
Pictures obtained by transmission rlectron microscopy (TEM) of a cell of *U. compressa* from a control site (Cachagua, **A**–**D**) and a copper-polluted site (Caleta Palito, **E**–**H**) at different magnifications. Arrows indicate electrodense nanoparticles (EDNP) that do not contain copper ions (-Cu) and that contain copper ions (+Cu). Cell wall (CW); starch granules (SG); pyrenoid (P) and chloroplast (CH).

**Figure 6 ijms-22-10531-f006:**
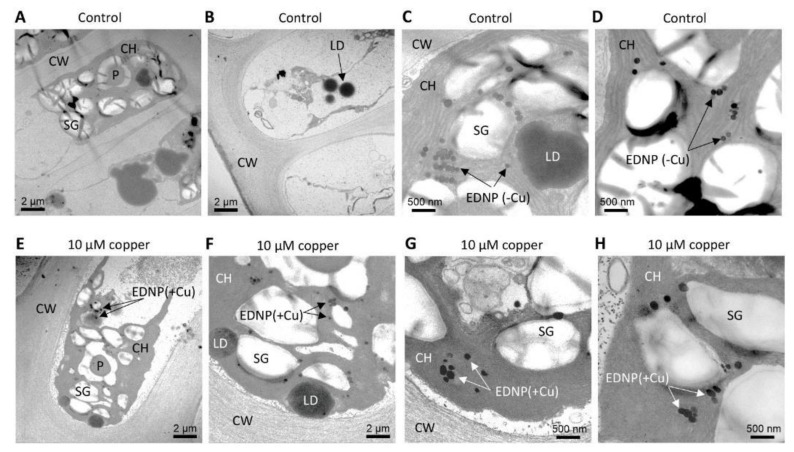
Pictures obtained by transmission electron microscopy (TEM) of a cell of *U. compressa* from a control site cultivated without copper addition in seawater (Cachagua, **A**–**D**) and with 10 µM of copper for 5 days (**E**–**H**) at a different magnification. Arrows indicate electrodense nanoparticles (EDNP) that do not contain copper ions (-Cu) and that contain copper ions (+Cu). Cell wall (CW); starch granules (SG); pyrenoid (P); lipid droplet (LD) and chloroplast (CH).

**Table 1 ijms-22-10531-t001:** Levels of copper, cadmium and lead in seawater of control sites and copper-polluted sites in northern and central Chile.

	Copper (µg L^−1^)	Cadmium (µg L^−1^)	Lead (µg L^−1^)
Zenteno	nd	nd	nd
Bahía Inglesa	nd	nd	nd
C. Aceituno	nd	nd	nd
Ventanas	nd	nd	nd
Cachagua	nd	nd	nd
Caleta Palito	1.97	nd	nd
Chañaral Sur	3.62	nd	nd

nd: not detected < 0.19 μg L^−1^.

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
