# Peer review of "Ulva compressa from Copper-Polluted Sites Exhibits Intracellular Copper Accumulation, Increased Expression of Metallothioneins and Copper-Containing Nanoparticles in Chloroplasts"

_ijms, 2021, doi:10.3390/ijms221910531_

Round 1

Reviewer 1 Report

Dear Editor/Authors,

The manuscript by Espinoza et al. is well in the scope of the journal and it may contain novel findings that can help to to analyze the mechanisms involved in copper accumulation in Ulva compressa. However, I have two critical questoins:

The aim of the manuscript should be added, the authors at the end of the chapter presented what they managed to achieve, so you can move this part to the conclusion - as an elaboration of the conclusions.

Should be also added to the introduction a few sentences about the use of Ulva compressa  in industry. Which would additionally justify the importance of this research.

Author Response

Dear reviewer: a sentence was added at the end of the introduction to highlight the aim of the work; conclusions were deleted in this paragraph, and  a sentence was added to explain the importance of U. compressa as a heavy metal biomonitoring species all over the world. All sentences are highlighted in red. 

Reviewer 2 Report

The manuscript of Espinoza et al. presents interesting research on ecophysiological and biochemical response of Ulva compressa to chronic pollution of sea water with copper. The manuscript is very clearly prepared, data presentation is good and conclusions mostly supported by the results.

I propose few minor corrections in the manuscript:

  1. Please put attention to use Italic font when gene names are mentioned, eg. in lines 19, 24, 163, graphs and so on.
  2. Line 28 - it should be “their” not “thr”.
  3. Please write clearly what is the reason of copper pollution in Caleta Palito and Chanaral Sur locations?
  4. Line 153 – please use “markedly” instead of “significantly” sine observed difference is not statistically relevant for 4 from 5 control conditions.
  5. Line 177 – it should be “nanoparticles accumulate in algae…”.
  6. Figure 5 – Scale bars are not clearly visible.
  7. Line 228 – it should be “nanomolar”.
  8. Line 235 – please do not discuss unpublished data, it is not allowed. Add the missing results or remove the statement from the manuscript.
  9. Line 345 – it should be “total”.
  10. Line 346 and 350 – it should be “volume”.
  11. Line 347 – it should be “water”.
  12. Line 347 – it should be “water-DEPC”.

Author Response

Dear reviewer: changes you requested were performed and some of them are highlighted in red. The scale bar was added to TEM photographs and transcripts are now in italics. Thank you

Reviewer 3 Report

The review on the publication by Espinoza et al. under the title Ulva compressa from copper-polluted sites exhibits an increase in the level of phytochelatin PC4, expression of metallothioneins and accumulation of copper-containing nanoparticles in chloroplasts.

Below are my minor comments to the publication:

Line 104-111 This is not a proper place for results. This is an introduction part! ‘Algae of copper-polluted sites showed the accumulation of intracellular copper, an increase in the level of transcripts encoding UcMT1 and UcMT2 whereas no increase in UcMT3 level. Moreover, algae of copper-polluted sites showed copper-containing electrodense nanoparticles, in chloroplasts, and algae from control sites also showed electrodense nanoparticles but they did not contain copper. Futhermore, algae form a control site cultivated with copper showed nanoparticles that contain copper in chloroplast and these copper-containing particles were not present in algae cultivated without copper addition’

Line 108 Please change from ‘Futhermore’ to ‘Furthermore’

Comments to the whole publication: I think it’s more than enough to use ‘Transmission electron microscopy in one place and then change to TEM. Revise in whole manuscript. 

Author Response

Dear reviewer: changes you requested were performed. Thank you

Reviewer 4 Report

The manuscript reports the ecological effects of the high concentration of heavy metals, especially copper, in Chilean coastal areas. In particular, the authors report the case of the compressed Ulva algae, and evaluate the molecular effects. Copper accumulates in tissues and in particular in the plastidial photosynthetic apparatus. These results are not observed in algae grown in copper-poor environments, and laboratory experiments with excess copper confirm the accumulation of heavy metal in chloroplasts.

The manuscript is clear, interesting and with important ecological applications.

In particular, I would suggest that the authors better highlight the potential ecological applications of their data. Can their measures be applied in assessing the effects of heavy metal accumulation? Were these data observed only for copper or do other heavy metals also show this action? Are there any effects on the photosynthetic efficiency of the alga?

Author Response

Dear reviewer:  a sentence was added to introduction to highlight the importance of the alga as a heavy metal bio-monitoring species (highlighted in red).

Reviewer 5 Report

This is an interesting article about copper accumulation in marine alga Ulva compressa. The article reads well, although there are some misspellings and badly structured sentences. So, overview by a native English speaker is needed. All abbreviations should be explained, when they first appear in the text and total list should be included. 

It might be helpful to mark copper polluted sites and data from polluted sites (e.g. by an asterix) in the figures.  The sequestration of copper containing nanoparticles in Ulva suggests that this alga could be used in phyto-remediation of polluted sites (by harvesting copper containing algae). Further, algae used in food industry should be tested for copper.     

Author Response

Dear reviewer: a sentence was added to introduction indicating that U. compressa has been used a sentinel species for heavy metal contamination all over the world. Other requested changes were performed. Thank you